# Changes in Management Lead to Improvement and Healing of Equine Squamous Gastric Disease

**DOI:** 10.3390/ani13091498

**Published:** 2023-04-28

**Authors:** Lieuwke C. Kranenburg, Simone Henriëtte van der Poel, Tim Sebastiaan Warmelink, David Anthonie van Doorn, Robin van den Boom

**Affiliations:** Department of Clinical Sciences, Faculty of Veterinary Medicine, Utrecht University, Yalelaan 112, 3584 CM Utrecht, The Netherlands

**Keywords:** equine, horse, stomach, ulcers, gastric ulcers, equine gastric ulcer syndrome (EGUS), equine squamous gastric disease (ESGD)

## Abstract

**Simple Summary:**

Stomach ulcers are common in horses. They cause pain and are a serious welfare problem. Several risk factors have been identified and stomach ulcers are routinely treated with omeprazole, a drug that reduces stomach acid secretion. Fourteen mares with severe stomach ulcers, previously used as embryo recipients, were studied. The horses were kept in individual stalls, exercised in a horse walker once a day and fed ad libitum hay and a small amount of low starch hard feed. Half of the horses were also fed a supplement containing hydrolysed collagen. After 4 weeks, the severity of the ulcers had decreased in both groups and ulcers had healed in 7 out of 14 horses. There was no difference between the horses that had received the supplement and those that had not. Severe stomach ulcers can improve, and even heal, with a diet of ad libitum forage, without the use of omeprazole. A predictable daily routine, with a limited number of dedicated caretakers, may have contributed to the improvement of gastric health.

**Abstract:**

Equine squamous gastric disease (ESGD) is common in horses and poses a serious welfare problem. Several risk factors have been identified and ESGD is routinely treated with omeprazole. Fourteen mares, previously used as embryo recipients and diagnosed with ESGD, were selected. Horses were confined to individual stalls, exercised once daily, and fed ad libitum hay, 1 kg of a low starch compound complementary feed and a mineral supplement. Half of the horses received a compound containing hydrolysed collagen (supplement) and the other half did not (control). At the start of the study, ESGD scores were 3.57 and 3.36 for the supplement and control group, respectively. After 4 weeks, the ESGD grades were significantly reduced in both groups (1.89 and 1.43, respectively, *p* < 0.01), and healing (ESGD < 2) occurred in 7 out of 14 horses. No treatment effect was observed (*p* = 0.75), and it was concluded that the change in husbandry overshadowed any potential effect of the compound. Severe ESGD can improve, and even heal, with the provision of a diet of ad libitum forage and a small amount of a compound complementary feed, without the use of omeprazole. A predictable daily routine, with a limited number of dedicated caretakers, may have contributed to the improvement of gastric health.

## 1. Introduction

Equine gastric ulcer syndrome (EGUS) is the term used to describe ulcers in a horse’s stomach; however, a distinction should be made between equine squamous gastric disease (ESGD) and equine glandular gastric disease (EGGD) [1]. Equine squamous gastric disease (ESGD) is common in both feral [2] and domesticated horses [3,4,5], with the highest prevalence in racehorses [6]. It poses a serious welfare problem, although ulcers can also be present without causing symptoms, and many horses with gastric ulcers, identified by gastroscopy, are considered asymptomatic [5,7]. However, Murray et al. [8] showed that gastric ulcers in adult horses can be clinically significant, with ulceration being more severe in horses with clinical signs. The fact that ESGD is more common in domesticated than in feral horses [2] highlights the importance of husbandry factors in the development of ulcers.

Several risk factors, mainly dietary and husbandry practices, have been identified. Animal husbandry is defined as the controlled cultivation, management and production of domestic animals, including improvement of the qualities considered desirable by humans by means of breeding. It includes day-to-day care, housing and exercise routines.

Diet plays an important role in both ESGD and EGGD, which can be induced by feeding high starch, low fibre diets [9,10]. Specifically, feeding more than 2 g of starch per kg of bodyweight per day or more than 1 g of starch per kg of bodyweight per meal has been associated with an increased risk of EGUS [7]. Easily fermentable carbohydrates lead to the production of volatile fatty acids, and in vitro studies have demonstrated that these have a negative effect on the gastric mucosa [11,12]. Increased periods of fasting because of a lack of forage opportunity also increase the risk of ESGD [7,13,14]. Changes in diet may also be important, as Chameroy et al. [15] found that only a small number of horses used in a university riding program, and not subjected to such changes, had squamous gastric ulcers, while these are common in other horse populations [15].

The transport of horses led to an increase in the severity of gastric ulceration, possibly related to the stress that this induced [16], while a low prevalence (11%) of ESGD was found in a group of university riding horses that never travelled [15].

Stress is often cited as a risk factor for EGUS [5,17], but scientific proof is scarce. Cribbing and other stereotypies are associated with ESGD and considered to be coping mechanisms, suggesting a link with stress [18,19]. Daniels et al. found no difference in gastric anatomy and physiology between crib-biting and non-crib-biting horses and suggested that there is no direct link between crib-biting and EGUS, but that both may be related to physiological and environmental stress [20]. It has also been demonstrated that horses with EGUS experience more oxidative stress than healthy horses [21]. Indirect support for the role of stress in ESGD was provided by the finding that the hair cortisol concentration was inversely related to the severity of ESGD [22], and it was postulated that high plasma cortisol concentrations associated with stress caused lower hair cortisol through a negative feedback mechanism. Moreover, the temporary relocation of horses was associated with a change in hair cortisol concentrations [23]. Both the trainer and the number of caretakers were shown to influence the prevalence of EGGD in riding horses; further, dealing with certain or many different people may be stressful for some horses [24]. The same may apply to ESGD.

Stall confinement was found to cause gastric ulceration, even when ad libitum hay was provided [25], and 55% of dietary-induced squamous ulcers healed when horses were turned out to pasture [26]. However, horses that are turned out to pasture can also be affected by EGUS, and Tamzali et al. [4] even found a higher gastric ulcer severity score in endurance horses kept only at pasture, compared to those turned out for part of the day. Le Jeune et al. [27] found that 71% of broodmares kept at pasture had gastric ulcers, with the majority present in the squamous mucosa. The mares in that study were seen to be grazing for approximately 16 h per day, fed hay and approximately 1 kg of a compound complementary feed per day, and kept in herds of 5–21 horses.

Very few squamous ulcers heal spontaneously, especially when horses remain in (race) training, and most require medical treatment [28,29]. However, spontaneous healing can occur, as demonstrated by the fact that three of eight vehicle-treated control horses showed ulcer resolution in 14–25 days, although healing occurred more often and more rapidly in horses treated with omeprazole [30]. However, because of the impression that spontaneous healing does not occur, or very rarely, many studies investigating the efficacy of various treatments have not included (placebo) control groups, because this is considered unethical [31]. This may have maintained the notion that spontaneous healing is rare and contributed to the reliance on proton pump inhibitors, mainly omeprazole, in the treatment of ESGD, and an underestimation of the importance of various husbandry practices.

Due to the costs of omeprazole; the fact that it is banned by some equine sport governing bodies; as well as the desire, among owners, trainers and vets, to limit the use of medication as much as possible, other methods of treating and/or preventing ESGD have been investigated. Moreover, (long-term) omeprazole therapy may not be (as) harmless as has long been thought [32].

Hydrolysed collagen (HC) has received considerable interest because of its gastroprotective, anti-inflammatory and cutaneous wound healing effects [33,34,35]. Properties that could make HC suitable for the treatment and/or prevention of ESGD are its enhancing effects on wound healing and combatting inflammation, as well as its suggested antioxidant properties [35,36]. Ingestion of HC results in the release of short glycine- and proline-containing peptides into the bloodstream [37,38]. Mucosal administration of HC peptides resulted in decreased gastric inflammation and reduced circulating amounts of the pro-inflammatory cytokine IL-1β in rats with stress-induced ulcers and acetic acid-induced ulcers [34,37]. In stall-confined horses undergoing intermittent feeding, the long-term (56 days) feeding of HC led to fewer non-glandular ulcers than in the control group [39]. The aim of the current study was to investigate the effect of a feed supplement containing hydrolysed collagen on the healing of ESGD in horses.

## 2. Materials and Methods

### 2.1. Animals

Ethics approval was obtained from the Animal Ethics Committee of Utrecht University (AVD1080020185204-3-01). Mares that had been used by private horse breeders as embryo recipients were returned to the Department of Clinical Sciences at Utrecht University and used for the current study. Prior to being enrolled in the study, the mares were kept at pasture in groups and fed additional forage as necessary.

A first selection of horses was based on an acceptable temperament while being handled, a body condition score (BCS) of 4 or 5/9, normal findings during a general physical examination and suitable behaviour while being exercised in a horse walker. During this first selection process, five horses were excluded from the study. The 28 remaining mares were weighed using a weigh bridge and underwent gastroscopy to assess the presence and severity of equine squamous gastric disease (ESGD), as reported by Sykes et al. [1]. Two mares were deemed unsuitable for enrolment in the study due to a recently diagnosed pregnancy and delayed gastric emptying, respectively.

The 14 most severely affected horses were selected and stratified according to ESGD grade and anatomical distribution of gastric ulcers. Within each pair, one horse was randomly selected to receive a feed supplement containing hydrolysed collagen (HC) and the other horse was assigned to the control group. During the first gastroscopic examination, four horses presented with *Gastrophilus intestinalis* larvae and were dewormed with ivermectin paste (Equimectin, 0.2 mg of ivermectin/kg) before the onset of the study. One horse presented with skin lesions resembling a fungal infection and was treated with a product containing enilconazole (Imaverol, 100 mg/mL). During the study itself, no medication was administered.

### 2.2. Gastroscopic Examination

Gastroscopic examinations were performed by one of the authors (LCK) as part of the selection process, over two days directly before onset of the study and immediately after the 28-day study period, on Day 29. To ensure maximum visualization of both the squamous and glandular mucosa, all horses were muzzled 12 h prior to gastroscopy, after the last evening feed on Day −1 and Day 28, and feed was withheld during this period. Horses were sedated with detomidine (Domosedan, 0.01 mg/kg bwt, i.v.) and gastroscopy performed using a 3.3 m long, 25–50 mm gastroscope (Storz telecam SLII 20213020). During the examination, the margo plicatus and the lesser curvature, the pylorus, antrum and fundus were inspected to assess gastric mucosal health. To enable observation of these anatomical regions, the stomach was insufflated using an insufflator (Storz CO2mbi LED TL100) and any remaining feed material and mucus were rinsed from the mucosa with tap water, as necessary. The entire procedure was recorded on video, and photographs were obtained of the margo plicatus, the lesser curvature and the pylorus and glandular mucosa. Lesions were graded as previously reported [1].

### 2.3. Experimental Design

The horses enrolled in the study were confined to 3 × 3 m individual stalls two days before the start of the study and for its duration, but they were able to see, smell and hear other horses at all times. All horses were exercised in a horse walker for one hour every day. Grass hay was provided ad libitum to all horses and water was available via paddle-operated automatic waterers in all stalls. A compound complementary feed was provided to all horses, starting on D1, twice daily, in equal portions in the morning and evening (08:00 a.m. and 20:00 p.m.). Horses in the treatment group received a total of 1 kg of a commercial compound complementary feed (Voermeesters Basis 10 mm, provided by Voermeesters B.V., Lienden, The Netherlands), containing 198 g of starch, every day, plus 100 g of the HC supplement pelleted into the compound complementary feed (provided by Voermeesters B.V., Lienden, The Netherlands). Horses in the control group received only 1 kg of the commercial compound complementary feed. Additionally, all horses were fed 600 g of a mineral feed (Subli Mineralenbikkels 10 mm, provided by Voermeesters B.V., Lienden, The Netherlands), containing 57 g of starch, once a day in the afternoon (13.00 h). The macronutrient content of the hay, complementary compound feed and mineral feed are presented in Appendix A.

All horses were observed and examined daily by two of the authors (SHvdP and TSW), who also fed the horses and took them to and from the horse walker throughout the entire study. This protocol was maintained for 28 days, after which gastroscopic examination was repeated on Day 29. The body condition scores were also assessed again, and the horses were weighed using a weigh bridge.

The photographs and videos taken during the gastroscopies were reviewed and ESGD graded by two diplomats of the European College of Equine Internal Medicine (LcK, RvdB), who were blinded to the treatment group. The mean scores of the two observers were used, which resulted in continuous data for each horse and each day.

### 2.4. Statistical Analysis

The ESGD grades are reported as means ± s.d., and differences between the supplement and control group were investigated using a Wilcoxon rank sum test on the matched pairs. Next, the ESGD grades in the supplement group on Day 0 and 29 were compared using a separate paired *t*-test, and the same for the control group. Statistical analyses were performed using RStudio (Posit software, Boston, MA, USA) and differences considered significant when *p* < 0.05.

## 3. Results

Of the 26 mares screened gastroscopically before inclusion in the study, 23 (88%) were diagnosed with ESGD ≥ grade 2. The mean (± standard deviation) bodyweight of the 14 selected mares at the start of the study was 565 ± 40 kg (range: 489–624 kg). The horses consumed 255 g of starch per day: 198 g in the compound complementary feed and 57 g in the mineral feed, which equates to an average daily intake of 0.45 g of starch/kg BW (range: 0.41–0.52 g/kg BW).

No significant treatment effect was observed between the supplement and control group on Day 29 (Wilcoxon rank sum test, *p* = 0.75, Figure 1). When only ESGD ≥ 2 lesions were taken into account, no significant difference in the number of ESGD ≥ 2 lesions could be observed between groups on Day 29. Of the horses included in the study, the mean starting ESGD scores (D0) were 3.57 and 3.32 for the supplement and control group, respectively (Figure 2). After 4 weeks of treatment (D29), the mean ESGD score in the supplement group had decreased to 1.89 (*p* < 0.001, 95% confidence interval: 0.850–2.507, mean improvement: 1.68), while the mean ESGD score in the control group was reduced to 1.29 (*p* < 0.001, 95% confidence interval: 1.194–2.877, mean improvement: 2.04).

The bodyweight increased in all mares during the study, to a mean of 595 ± 36 kg (range: 515–646 kg), while there was no discernable change in BCS.

After the 28-day treatment period, 13 of 14 horses (93%), of which six received the supplement and seven received the normal diet, had lower ESGD scores on day 29 than at the start of the study on Day 0 (Table 1). The remaining horse, which received the supplement, did not show any improvement but rather a slight increase in lesion grade (horse 02, Table 1). The individual ESGD scores for each horse before and after treatment are displayed in Table 1. Only horses with clinically relevant ESGD lesions (ESGD0 ≥ 2) were selected for this study. Thus, on Day 0, all 14 horses presented with ESGD lesions ≥ 2. On Day 29, seven horses presented with ESGD lesions < 2 and had therefore healed (Table 1). Of these seven horses, four were in the control group and three in the supplement group.

## 4. Discussion

In the present study, the high prevalence (88%) of ESGD found in the mares that had been kept at pasture before the study began is comparable to that seen in racehorses in training [6]. Turnout at pasture is often considered ideal and, in a study looking at the prevalence of ulcers in equids in a safari park, the highest numbers were found in those animals that were stabled at night, as opposed to those turned out permanently [40].

In the current study, an improvement in ESGD score was observed in nearly all horses over 4 weeks, after they were moved from pasture to individual stalls, and healing occurred in 7 of 14 animals. This is somewhat unexpected, as spontaneous healing is considered uncommon, but similar to what was reported by Woodward et al. [41], although the reduction in ulcer scores at 21 days was only significant in the control group in that study. As there was no difference in terms of the improvement or healing of ESGD between the supplement and control groups, the explanation must be sought in other factors. The effect of the management changes overshadowed any potential effect of the HC supplement.

Before entering the study, the mares were kept at pasture and able to graze, and they were fed supplemental haylage at different feeding stations. During the trial, the horses were fed ad libitum hay and approximately 1 kg of compound complementary feed per day. Ad libitum roughage is crucial in the prevention of ESGD, while feeding more than 1 g/kg of starch per meal or more than 2 g/kg of starch per day has been shown to be a risk factor for EGUS [7]. Neither diet seems likely to cause gastric ulcers, provided horses at pasture all had equal access to the supplemental roughage. It is possible that some mares, because of their position in the herd hierarchy, were prevented from eating haylage, leading to feed stress. The importance of a low-starch diet in the prevention of ESGD was demonstrated by Luthersson et al. [42], who showed that ESGD recurred in horses with severe ESGD, following improvement while being treated with omeprazole. In the current study, the diet of the mares before they returned to the University was unknown, and it is possible that they had been fed high-starch diets, as is common for brood mares. During the study, the horses consumed 0.45 g of starch/kg bodyweight (on average), which is lower than the maximum of 1 g/kg of BW/meal or 2 g/kg of BW/day advised by Luthersson et al. [7] and lower than when feeding traditional high-starch sweet or pelleted feeds [43]. After returning to the University, the horses did not receive compound complementary feed prior to entering the study. Another risk factor for EGUS is the unavailability of water in the turnout paddock [7]. Automatic waterers were present in the paddocks and in the stalls, but again, access may have been limited for some horses at pasture due to the herd hierarchy; however, this seems less likely for water than for feed, as equids do not drink often and mainly in the periprandial period [44].

Stress is considered to play a role in the formation of both squamous and glandular ulcers in horses [5,16,17,18,19,22,45,46,47], although it can be difficult to determine if or when horses experience stress and to quantify its degree and duration. It is often thought that being housed at pasture with other horses is ideal, as that is how feral horses live in the wild. However, in the wild there are no fences around the pasture and horses can escape stressful situations. Different horses have different temperaments and behavioural responses to stimuli [48,49], and hierarchies exist within groups of horses [50]. Contact with other horses may be stressful for (some) horses, especially those that are low in the hierarchy, and unstable social grouping increases aggression and may lead to injury [51,52].

It has been shown that horses can react very differently to novel experiences, and some show a very obvious stress response, including tachycardia and a flight reaction [48,49]. It seems likely that (some) horses will also react with a stress response to unpredictability or variations in their daily routine. In the current study, the daily routine was the same every day, with set feeding times and exercise in the horse walker at the same time of day. It was observed that horses displayed calmer behaviour as the study progressed, suggesting they became accustomed to this daily routine.

Both the trainer and the number of caretakers were shown to influence the prevalence of EGGD in riding horses [24], indicating the importance of the human–horse interaction, which may also lead to stress. This may also be related to predictability and is likely dependent on the caretakers’ experience in handling horses. In the current study, the horses were cared for exclusively by two of the authors (SHvdP and TSW), both of whom are very experienced in handling horses, and the mares’ behaviour seemed to change during the course of the study, generally becoming calmer. This change could be related to the interaction between the caretakers and the horses, as the latter responded clearly to the arrival of the caretakers in the stables and habituation to the new routine. It could also be related to the change in diet after moving from groups at pasture to indoor stables.

This study has a number of limitations, including the fact that it was not (specifically) designed to investigate the effects of changes in husbandry, nor to assess (changes in) the horses’ behaviour. In future studies, the behaviour or types of behaviour displayed by the horses, such as foraging, locomotion, interaction with other horses and/or caretakers, should also be studied. Moreover, the degree of stress experienced by horses could be quantified by measuring cortisol concentrations in hair, for example [22]. However, our findings emphasise the importance of including a control group when studying the effects of interventions on gastric ulcers in horses. This is not always the case, as the spontaneous healing of ulcers is thought to be rare, and it may be considered unethical to withhold treatment in horses with EGUS [31]. Moreover, it is hard, or maybe even impossible, to separate changes in diet, or how rapidly or gradually these are introduced, from other husbandry practices. When horses were moved from pasture to individual stalls their diet changed from grazing, supplemented with haylage, to ad libitum hay and a small amount of compound concentrate feed. Neither diet would be expected to cause gastric ulcers, although it cannot be excluded that some horses were not able to eat ad libitum roughage at pasture, prior to the start of the study, as a result of the social structure in the herd.

Finally, stress is thought to contribute to EGUS development, but no attempt to quantify stress was undertaken in the present study. Stress may be quantified by measuring cortisol concentrations and, in relation to EGUS, it would seem logical to assess stress, and therefore cortisol concentrations, over a longer period. Prinsloo et al. [22] reported an inverse correlation between hair cortisol concentration and the severity of ESGD, which was attributed to a negative feedback mechanism, with higher serum cortisol leading to lower hair cortisol concentrations. In another study, the hair cortisol concentration in police horses changed significantly 6 weeks after relocation, indicating that this timeframe is long enough for such changes to occur and that a change in management is associated with a change in stress levels [23]. In future studies, it would be worthwhile to study the behaviour of horses and measure hair cortisol concentrations before and after the treatment of gastric ulcers, in order to determine if a change in EGUS severity is correlated with a change in hair cortisol concentrations.

## 5. Conclusions

Severe ESGD can improve, and even heal, in 4 weeks with the provision of a diet consisting of ad libitum roughage and a small amount of a low-starch compound complementary feed, and without the use of proton pump inhibitors or other medication. A predictable daily routine, with a limited number of dedicated caretakers, may have contributed to lower stress levels in horses and the improvement of gastric health. Future studies should examine the effects of different husbandry practices on the occurrence and healing of gastric ulcers in horses.

## Figures and Tables

**Figure 1 animals-13-01498-f001:**
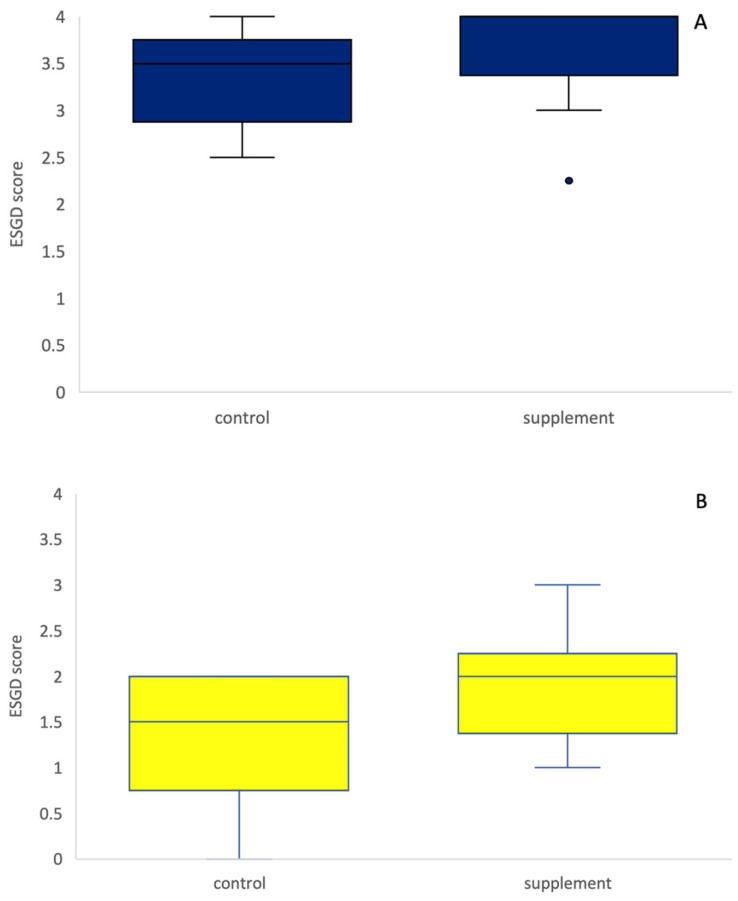
ESGD scores on Day 0 (**A**) and Day 29 (**B**) in the control (blue) and supplement (yellow) groups. There was no difference in the ESGD score between the control and supplement group (*p* = 0.75), neither on D0 nor on D29.

**Figure 2 animals-13-01498-f002:**
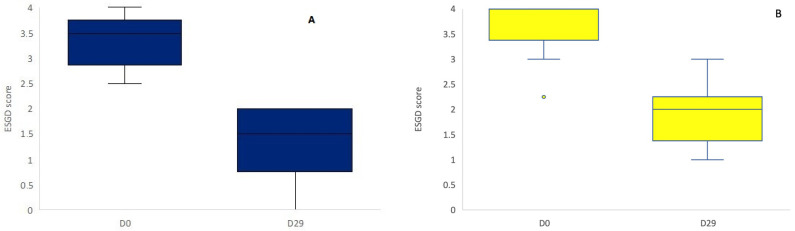
ESGD scores in the control group (**A**, blue) and the supplement group (**B**, yellow) on Day 0 and Day 29. There was a significant reduction in ESGD score on D29 in both the control and the supplement group (*p* < 0.001).

**Table 1 animals-13-01498-t001:** ESGD scores (mean of the two observers) in individual horses before (ESGD0) and after (ESGD29) the test period.

CONTROL			SUPPLEMENT		
HORSE ID	ESGD0	ESGD29	HORSE ID	ESGD0	ESGD29
01 *	2.5	0.5	02	2.25	2.5
03 *	4	0	05 *	3.75	1
04 *	3.5	1	08	3	2
06	3.25	2	09 *	4	1.75
07	2.5	2	10	4	3
12 *	4	1.5	11	4	2
13	3.5	2	14 *	4	1

After the treatment period, seven horses had ESGD grades < 2 (*). On Day 29, 7 horses had ESGD scores < 2 and were considered healed.

## Data Availability

All data are provided in this paper.

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
