# Peer review of "Changes in Management Lead to Improvement and Healing of Equine Squamous Gastric Disease"

_animals, 2023, doi:10.3390/ani13091498_

Round 1
Reviewer 1 Report
Dear authors,
I have revised the manuscript titled “Changes in management lead to improvement and healing of Equine Squamous Gastric Disease”.
Introduction
Line 49 – I suggest to explain better which “husbandry practices”
Line 51 – I have seen that within the references 9-11, a couple of them derived from in-vitro studies. I could suggest to refer to this recent study “Colombino et al. 2022 BMC Veterinary Research, doi 10.1186/s12917-022-03433-y “; in which the authors found more severe lesions on gastric mucosa in horse fed high starch compared to high forage diet.
Moreover, please define “high concentrate/starch diet” informing the reader about safe level of starch/kg horse BW.
Periods of forage deprivation … do you mean the increase in fasting time according to a lack of forage opportunity?
Line 93 – why there is this evidence? Moreover, I suggest to inform the reader what it is hydrolysed collagen and the mechanism of action
M&M
Where the study was carried out? In one reproduction farm?
Line 120 – please, specify who performed the gastroscopy and if the gastroscopy was always performed by the same person.
Line 135 – I would suggest to add the nutritional table of the diet fed to the horses.
Moreover, It is not clear which kind of concentrate feed – cereal concentrate? How many concentrate and starch/meal?
It is not clear if the horses were accustomed to the diet… if yes, in how many days you increased the quantities?
Lines 153-156 – I suggest to add a specific statistical analysis section
Results
Line 166 – what about BCS? Pay attention because the fullness of the digestive system could influence the BW … When the BW was measured at D0 and D29? And how BW was measured?
Line 178 – are displayed in table 1?
Discussion
Line 222 – check for reference formattation
Line 252-254 – I think that the problem is related to the gradually change of the diet and not to the change itself.
Lines 257– also the study of behaviours are important to determine the level of frustration and stress of horses.
Conclusions
In your conclusions you should focus on the main results of your study. In your conclusions (also in the simple summary and in the abstract) You can suggest to improve husbandry practices as daily routine and so on …. but you should only suggest since you did not test husbandry practices…
References
13 – authors are underlined.
Author Response
Please see document for an itemized response

Reviewer 2 Report
Dear Authors this is a nice study and a great example of not finding what you were looking for but still adding useful knowledge to the field.
There are a few pointers and comments to improve the manuscript prior to publication.
You use the term concentrate feed several times in the manuscript starting in lines 13, 24, 50, 75, 140 etc the more appropriate term in line with EU feed law is compound complementary feed.
Within the introduction you mention high starch diet as a risk factor for EGUS (line 51) but it would be useful to give a g/kg of starch to give the reader context on what is meant by high starch.
Line 53 the stress link i agree is difficult to pin down with EGUS and also with stereotypic behaviour. https://doi.org/10.1016/j.jevs.2019.102853 this study suggests that EGUS is linked to oxidative stress as suggested as a link between EGUS and crib biting from this paper https://doi.org/10.1016/j.jveb.2018.12.010
Method - line 141 onwards to 145 Given this is a nutrition study i am surprised that there is no nutrient profile for the diet. While im assuming this was a low starch compound complementary feed we dont know the starch content of the feed or the total diet, this is a consideration for interpreting the results. Similarly at pasture the horses had access to haylage, what was the difference in nutrient profile and how long were then on the hay diet before T0 screening?
Line 153 - Statistical analysis - this is not coming across clearly so some clarity is needed. Why use a T-test between D0 and D29 and then Wilcoxon between groups. You could argue that giving the score are on the EGUS scale which is not continuous and infinite then a non parametric approach should be used. My observation from the manuscript is that you are interested in : Between group pre-supplementation and between group post supplementation, or pre and post irrespective of group? Given this was an intervention study the statistics should really be repeated measures so Wilcoxon matched pairs T0 and then T29. Given the limit of non-parametric repeated measures ANOVA options it would seem logical to use Wilcoxon Pre supplemented and Pre control and another wilcoxon post supplemented and post control to look between groups. To correct for type one error then apply a Bonferroni correction. The data quite clearly show the reduction in the way they are presented but the method of analysis is currently unclear.
Line 172- figure 2 legend refers to diamonds, there are no diamonds.
Discussion covers the key points. i wonder if the works of Luthersson et al 2019 DOI: 10.1016/j.jevs.2019.05.007 where horses still were dosed with omeprezole but diet was altered to lower starch and improvement was seen may be useful?
Was there any interaction between the mares that presented with Gasterophilus pre trial and post trial EGUS scores?
Author Response

(The authors gave the same response as above.)

Round 2
Reviewer 1 Report
dear authors, the paper improved a lot, I endorse the pubblication
Author Response
Thank you!
Reviewer 2 Report
The manuscript has been greatly improved and actually this information is important to show the role of management in managing EGUS.
There are a couple of comments that I feel need to be addressed further:
Methods- mean weight of the animals at the outset would be useful, it is clear to see with the information on the starch intakes that they were on a low starch diet but I feel you need to make this very clear that low starch complementary compound feeds have a place but the starch intake per KG or BW need to be low as it is here.
Stats - I think you need to explain this a bit better as its taken me a few reads to understand. Initially for each group you did a matched pairs T test to compare ulcer grade between day 0 and day 29, fine, then you did a T test between groups, you could argue they were controls and not exact matched pairs but either way also fine. So you need to be clear initially you tested for changes in Ulcer score within groups whereby each animal acts as its own control (Pre-Post). Then you compared treatment and control group ulcer scores.
I would be inclined to present the other way around. Ive used your data and i did a Mann Whitney U test on the two groups using all the data e.g. irrespective of D0 and D29 just looking at differences between groups T&C and as you have presented there was no effect of treatment. I think that is result 1, no effect of supplement. Then result 2 is within groups using a matched wilcoxon where animals are there own controls ulcers scores significantly improved in both groups.
Results - as per stats comments - it would be logical to have 1 plot first with the two groups irrespective of time point showing no difference from supplementation. Then in the other two plots (1 or A) Control group Pre and control group post and then in 2 or B Treated group pre- D0 and post D29 so it is clear the differences are within groups but not between groups.
Finally for the conclusion and abstract - as highlighted above when talking compound complementary feed make the point this was very low starch - your study was less the 0.5g/kg/bw per day so it is very clear complementary feed to balance a ration is great but low starch.
A nice study with minor tweaks it will be ready for publication.
Author Response
We have made further changes based on the second review and hope these are satisfactory/sufficient. If you have any further queries regarding our paper we will be more than willing to address these.
Please see document for itemized response.
